# Spatial Patterns and Influencing Factors of Rural Land Commodification at Township Scale: A Case Study in Shijiazhuang City, North China

**Lin Fu and Junko Sanada ***

Department of Civil and Environment Engineering, Tokyo Institute of Technology, Tokyo 152-8550, Japan;
fu.l.aa@m.titech.ac.jp
* Correspondence: sanada.j.aa@m.titech.ac.jp

**Abstract:** The rapid spread of capitalism in rural areas has facilitated rural land commodification (RLC). While some scholars have studied RLC, few have analyzed its spatial characteristics. Taking Shijiazhuang city as a study area, this paper applies Moran's I method and spatial regression models to analyze township-scale RLC patterns and driving factors. The study investigates four common pathways of RLC: production-oriented farmland, tourism-oriented farmland, rural homesteads, and construction land commodification which are predominantly found in urban fringe areas. The distribution of RLC demonstrates positive spatial autocorrelation, characterized by spatial aggregation and polarization. Population, economic level, agriculture, and location conditions are identified as key drivers, and their specific mechanisms vary across development pathways. Future efforts should focus on ensuring balanced and coordinated RLC development in accordance with regional conditions and capacity, while also addressing the implications arising from the coexistence of RLC with rural aging and poverty.

**Keywords:** rural land commodification; rural transformation; land use; spatial pattern; China rural

## 1. Introduction

Land is a mirror of society [1]. Especially in the countryside, land as the spatial carrier of human production and living closely reflects the evolution of socioeconomic activities. In recent decades, rural areas around the world experienced many transformations in spatial environments, productive processes, and social arrangements. For one thing, technological advances released agricultural resources, leading to multifunctional development. For another, with markets and new values, urban interests further consumed rural land for residential, recreational, and investment purposes [2,3]. This two-fold driving force of advancement and commodification gave rise to rural land commodification (RLC), a major transformation trend across the globe [4–6].

The discussion on land commodification can be traced back to Polanyi's work on the great transformation. In 1957, Polanyi systematically expounded on the rise of market society, pointing out that influenced by the market economy, fictitious commodities such as land and labor are undergoing an inevitable process of commodification [7]. The land commodification in rural areas involves the change in production mode and the human–land relationship and also provides a new theoretical understanding of the pathway of rural development. RLC facilitates the growth of commercial agriculture, rural tourism, and real estate development, leading to more market-oriented activities [8–10]. These developments have positive impacts on the rural economy and employment [11]. Meanwhile, the negative social effect caused by land commodification has also attracted discussions, such as the land scarcity problem among small farmers and social differentiation [12–14]. In the current era of economic globalization, RLC has become an unavoidable trend in many countries. Rural areas will determine their benefits or losses from this trend based on their

capacity for innovation and adaptation [15]. Therefore, exploring the factors that influence rural commodification is crucial for understanding its spatial complexity and formulating appropriate rural development plans.

Due to the dual land system, access to land in rural China was limited within the village collective and independent from the capital market over a long time. The 'monopole on property rights' induced the rigidity of human–land relations, severely restricting diversified economic development [16]. In response to this problem, the Chinese government proposed a new rural land reform to promote the market-oriented allocation of land resources in 2014. Moreover, to help the capital accumulation in the disadvantaged area, the No. 1 central document of 2015 encouraged the capital flow into the countryside. This strategy sparked a wave of entrepreneurial activity in rural China and directly led to the market-led resourcing movement of rural land. In general, China's countryside is undergoing an unprecedented commodification transition. Under the background of actively promoting rural transformation, a scientific understanding of this phenomenon is of great significance for optimizing the rural land structure and improving the sustainability of rural development.

Currently, the research content of RLC mainly includes developing process [17–19], driving mechanism [20,21], and spatial pattern [22,23]. In mechanism research, existing studies mostly employed theoretical models such as social construction, actor-network, and creative construction to analyze the influencing factor of RLC in specific villages [24]. The systematic examination, particularly in terms of cross-comparing various development models, has not been studied. Moreover, these studies generally revealed the driving mechanism from the perspective of social force and government interference, neglecting the potential influence of macro socioeconomic background and external conditions. At the same time, research related to spatial patterns is mainly conducted at the country or province level. However, in the actual situation, RLC within counties still shows spatial differences due to the variety of resource endowments. There is an urgent need to analyze at the micro-scale and explore how this phenomenon occurs and spreads according to local conditions. The township is the most basic unit of social and economic organization in rural areas, township scale analysis of RLC can be used to reveal and guide the future optimization of rural development and transformation. To fulfill current research gaps, this paper takes Shijiazhuang City as a study case, based on spatial analysis methods and regression analysis model, quantitively analyzing the spatial pattern, interaction, and driving factors of RLC at the township level. The goals of this paper are to address three pressing questions: (1) How do RLC spatially distribute? (2) What is the key driving factor for RLC? (3) How do regional social and economic characteristics affect RLC? The article is structured as follows: Section 2 presents a literature review that investigates the development pathway and influencing factors of RLC; Section 3 provides details of the study area, data, and methods; Section 4 shows the main results; Section 5 summarizes the conclusions and policy implications.

## 2. Research Basis

### 2.1. Theoretical Models for Rural Land Commodification

The term "commodification" refers to the process by which goods and services are increasingly produced by capitalists for profit under the conditions of market exchange [25]. According to [26], commodification has three distinct elements: produced for exchange; conducted under market conditions; and motivated by the pursuit of profit. Space has material attributes and can be constantly manufactured through the practice of production and consumption [27]. When we regard land as a "commodity", land commodification then represents a space production process driven by the pressures and demands from capital investment and market preference.

RLC started with the spread of capitalist markets in rural areas [28]. With capitalized invasions, spatial production promotes rural industry change, and the consumption function of rural land emerges as the market influents more of rural society. Furthermore,

because of the differences in political-economic and socio-cultural perspectives, commodification models in rural areas differ from one society to another. In developed countries, influenced by the impact of counter-urbanization, the function of leisure, environment, and cultural preservation in rural areas gradually emerged, leading to a wide "commodities" transition among land, agricultural products, houses, and natural landscapes. Meanwhile, in developing countries with rapid urbanization, rural land is inevitably included in the urban and rural land development system and becomes a major area for urban expansion and industrial migration.

This study referred to related research and divided Chinese RLC into four classes according to the commodification elements (Figure 1). Among these, product-oriented farmland commodification (POFC) reveals a trend in expanded and industrialized production of agricultural products based on farmland. In this context, traditional agriculture products become commodities with consumption value and traditional production patterns exhibit the characteristic of expansion, intensification, and specialization [29]. Examples of this type include family farms, agricultural cooperatives and agro-industrial parks which involve large-scale land consolidation and extensive investment for new farming techniques and infrastructure. Tourist-oriented farmland commodification (TOFC) also relates to the shift in agricultural land value, but more particularly it imparts the leisure and tourist function to farmland and offers aesthetic and experiential consumption through creating and packaging rural farmland. In China, leisure farms that focus on picking or agricultural experiencing exemplify the main characteristics of this type. Homesteads commodification (HC) refers to remodeling rural houses for accommodation, catering, and travel use. In many cases, this type of commodification rests on the anti-urban and rural idyllic sentiment of urban residents, particularly when the connection is established between rural symbols and pristine lifestyles [17,30]. An example of this type includes homestays and rural inns. Construction land commodification (CLC) relates to the flow of urban investment into rural areas. In the context of urban–rural land use imbalance development, rural areas attract external capital for their cheap rents and high potential profit so that the rural construction land is increasingly marketed for the development of the secondary industry. In this context, the construction land adapting to the development of rural industries, such as family workshops and industrial parks are typical examples of this type.

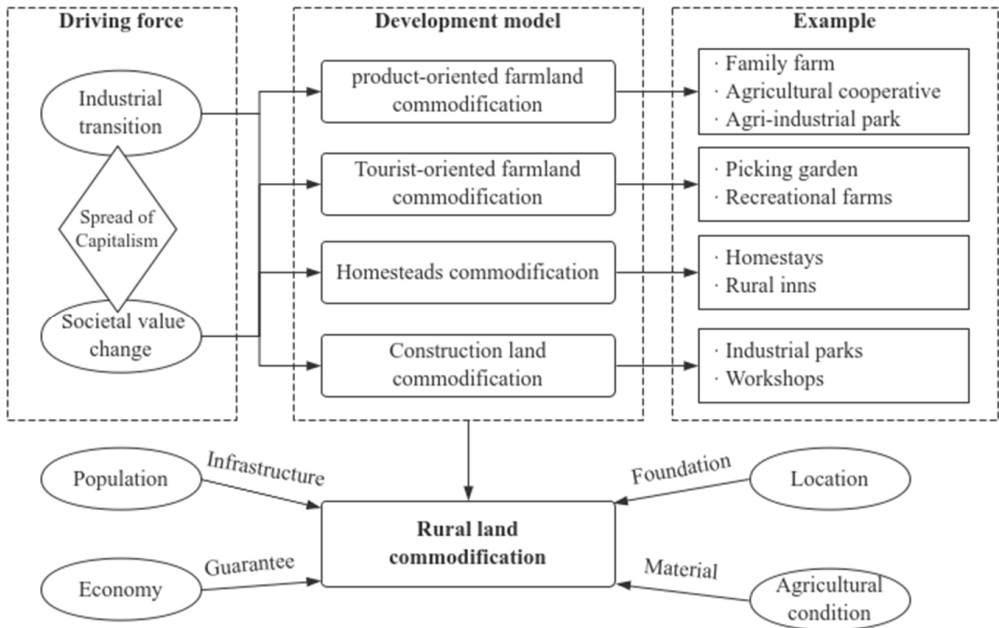

**Figure 1.** The theoretical framework of RLC.

*2.2. Theoretical Analysis of Driving Factors of Rural Land Commodification*

RLC is essentially a trend of rural transformation caused by the interaction of physical and socio-economic factors. The drivers of commodification can be likened to those of land use change, such as population, economy, agriculture, and location conditions [31]. To explain the commodification transition among rural land systems, it is important to factor in these elements.

Population factors are the basic condition of rural transformation and provide support for rural areas [32]. The driving force of population factors is mainly reflected in the human capital and labor force, which is usually measured by population density and demographic characteristics. A high population density provides sufficient labor resources and positively impacts farmland intensification and industry upgrading [33,34]. Population age is an important factor considered by capital investment [35]. The availability of working-age labor forces influences the investment intentions of developers and enterprises, which indirectly promote the land use transition in rural areas.

Economic condition is considered a guarantee for the process of rural transformation, including the level of economy and population income [36]. The GDP factors reflect the overall economic environment and to a large extent determine the investment flow and industrial construction in rural areas. Per capita income, as the key measurement of population living standard, not only influents farmers' choice about land use and livelihood but also affects the overall direction of rural development for its important indicator role in policy making and financial subsidies [37,38].

Location conditions affect the efficiency of people and material exchange between regions, and spatial differences in location result in different rural transformation characteristics. Several studies have found that good locality conditions can enhance rural industry transition and upgrading. For example, rural tourism is generally prioritized to occur in areas in proximity to urban areas and public transportation [31], and enterprises, as well as agricultural cooperatives, are more willing to invest in areas with physical accessibility [39].

Agriculture condition also has an impact on the transition of land structure. One study suggested the amount of arable land directly associate with the rural developing models; villages with sufficient arable land resources are prone to embark on an efficient and premium production pathway [40]. In addition, factors such as agriculture capacity and planting structure influence the allocation of agricultural resources, which in turn affects rural land use. Agricultural overcapacity leads to the release and transfer of agricultural resources into multifunctional use [41]. Monoculture planting structure, especially dominated by food crops, is less likely to embark on multifunctionality pathways [42].

## 3. Material and Method

### 3.1. Study Area

Shijiazhuang (SJZ) is the capital city of Hebei Province. It is about 266 km southwest of Beijing and consists of 22 counties with a total area of 15,850 km$^2$. The city stands at the edge of the North China Plain (Figure 2). The vast plain terrain provides conditions for large-scale agriculture, making it one of the most important agricultural production areas and densely populated rural settlements in China. Since 2018, Shijiazhuang had an agricultural population of 3.35 million and 3943 rural settlements, and its agricultural production scale even ranks first among Chinese key cities, providing sufficient research samples for the study of rural commodification. In addition, as one of the demonstration areas of Chinese rural land reform and rural revitalization policies, the rural areas in SJZ have undergone a tremendous transition in recent years, which can be regarded as a microcosm of rural China. Therefore, this study used SJZ as a typical case to study the spatial pattern of rural land commodification in specific cities.

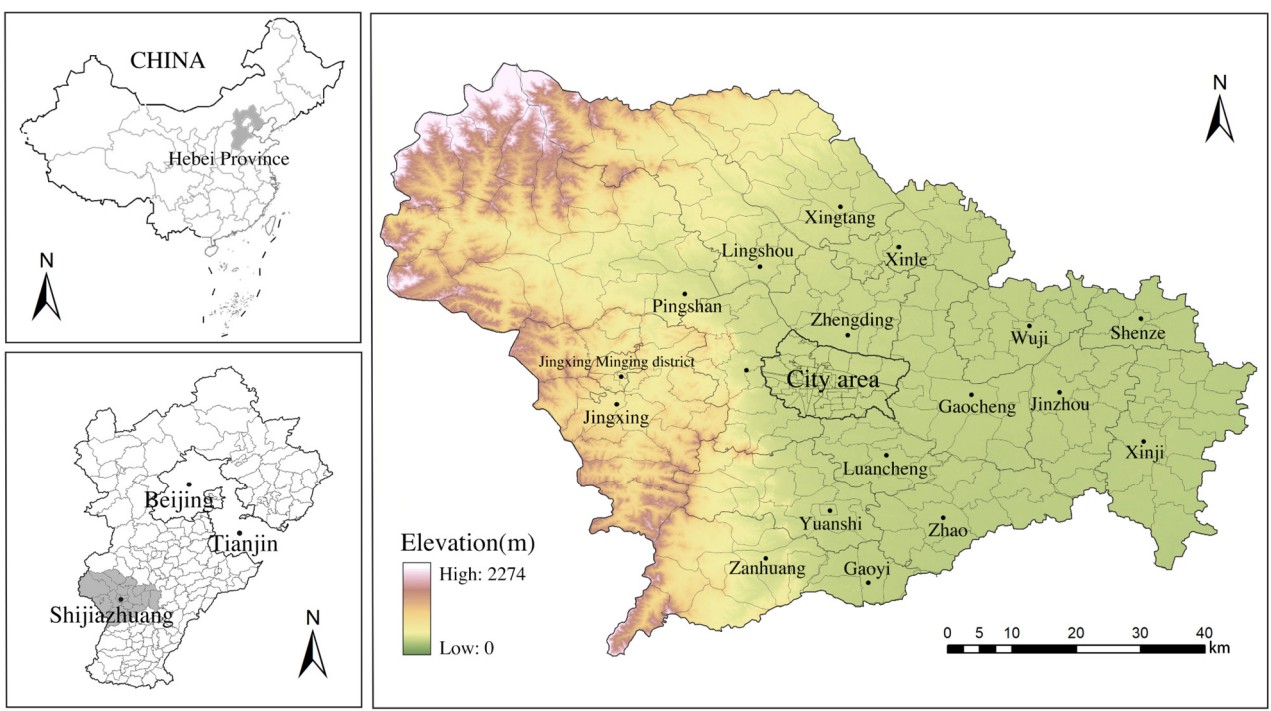

**Figure 2.** The study area of Shijiazhuang City.

*3.2. Data Sources*

This study took 209 administrative townships outside the urban area of Shijiazhuang City as the research unit. The data required in this paper included RLC representation data, agriculture and socioeconomic data, and basic geographic data.

(1)  This study took the point of interest (POI) data from online maps as the source of RLC representation data. AutoNavi Map (ANM) is one of the largest online map service suppliers in China. The POI data used in this study were collected from ANM in 2022 through the Application Programming Interface (API). Through the keyword searching, geographical points containing specific attribute fields such as name, category, type, and location were collected. Main searching keywords include 'family farm', 'agri-cooperative', 'agri-industrial park' for POFC, 'picking garden', 'leisure farm' for TOFC, 'rural homestay', 'rural inn' for HC, and 'rural industrial park', 'family workshop' for CLC. Furthermore, the Taobao village data provided by Ali Research (www.aliresearch.com) was also employed as supplement data of POLC and CLC, as its development is generally based on agricultural production or rural processing areas. To avoid duplicates and related data, these data were further checked manually to exclude the point value duplicates or located within the city area. In total, 3368 data points consisting of POFC (932 POIs), TOFC (668 POIs), HC (651 POIs), and CLC (1117 POIs) were obtained.

(2)  The agriculture and socioeconomic data mainly came from China Statistical Yearbook (Township), Hebei Rural Statistical Yearbook, and Hebei Township Economic Yearbook. The population data were from Chinese census data (2010). All the data above were obtained from the China Economic and Social Big Data Research Platform (http://data.cnki.net accessed on 25 May 2023).

(3)  The basic map data including the boundary of county and township districts were sourced from The Earth Science Data Sharing Centre at The Institute of Geographic Sciences and Natural Resources Research.

*3.3. Methods*

The following methodology was employed to analyze the spatial pattern and driving factors of RLC:

(1) The POI Data aggregated at the township level was used to produce the density map and spatial statistics of RLC. As a visual approach to the distribution, it measures the overall degree of land commodification, allowing the spatial patterns between different types of RLC to be compared.

(2) Based on aggregated data, spatial autocorrelation analysis was used to describe the distribution pattern and identify the spatial clusters of RLC. This study applied the Global Moran's I to evaluate spatial correlation, discerning whether commodification phenomena are distributed in clustered or dispersed patterns. The value is between negative one and one, wherein the value closer to one means that townships with similar RLC conditions are aggregation distributed and the value closer to negative one implies opposition. Anselin Local Moran's I(LISA map) was used to visualize clusters and outliers. Through local Moran's I analysis, we can categorize the distribution into four types: High-High clusters (a high value surrounded primarily by high values), Low-Low clusters (a low value surrounded primarily by low values), High-Low (high values surrounded primarily by low values) or Low-High (low values surrounded primarily by high values).

(3) Multiple regression models (OLS) were used to access factors influencing RLC. Based on the theoretical analysis in Section 2, this study took the RLC point density as the dependent variable. Furthermore, 10 independent variables were selected according to the available of data from the four categories of population, economy, agriculture, and location conditions (Table 1).

**Table 1.** Indicators for analyzing the driving factors of RLC.

| Category | Symbol | Variables | Definition |
|---|---|---|---|
| Population | $X_1$ | Total population | The number of the township population |
| | $X_2$ | Aging population rate | The proportion of the elderly population |
| Economy | $X_3$ | GDP per capita | The per capita GDP of township |
| | $X_4$ | Net income per capita | The per capita net income of farmer |
| Agriculture condition | $X_5$ | Cultivated area rate | Cultivated land area of township |
| | $X_6$ | Gross output | Total agricultural output value |
| | $X_7$ | Grain production output | Total grain production value |
| Location | $X_8$ | Dis to city | Distance from township to city center |
| | $X_9$ | Dis to county | Distance from township to nearest county |
| | $X_{10}$ | Dis to tourist spot | Distance from township to nearest tourist spot |

## 4. Results and Analysis

### 4.1. Distribution of Rural Land Commodification

The density map reflects the overall distribution and development level of RLC. In Figure 3, the density of the RLC data point at the township was calculated and divided into five grades by the natural break method, which includes very high, high, medium, low, and very lower. In general, the four types of RLC presented a clear core-periphery pattern with the very high-density areas distributed around the city centre and respective advantage area. Additionally, the density distribution of RLC also exhibits a strong polarization pattern. It is evident that among the four types of RLC, the proportions of high-density and very high-density areas only account for 19%, 6.2%, 11.4%, and 15.8%, respectively, with none exceeding 20%. RLC tends to develop intensively within a small range, reflecting the key controlling role of regional conditions on the commodification transition. Under the

combined influence of geography, policy settings, and economic levels, RLC development demonstrates an imbalance and polarization pattern.

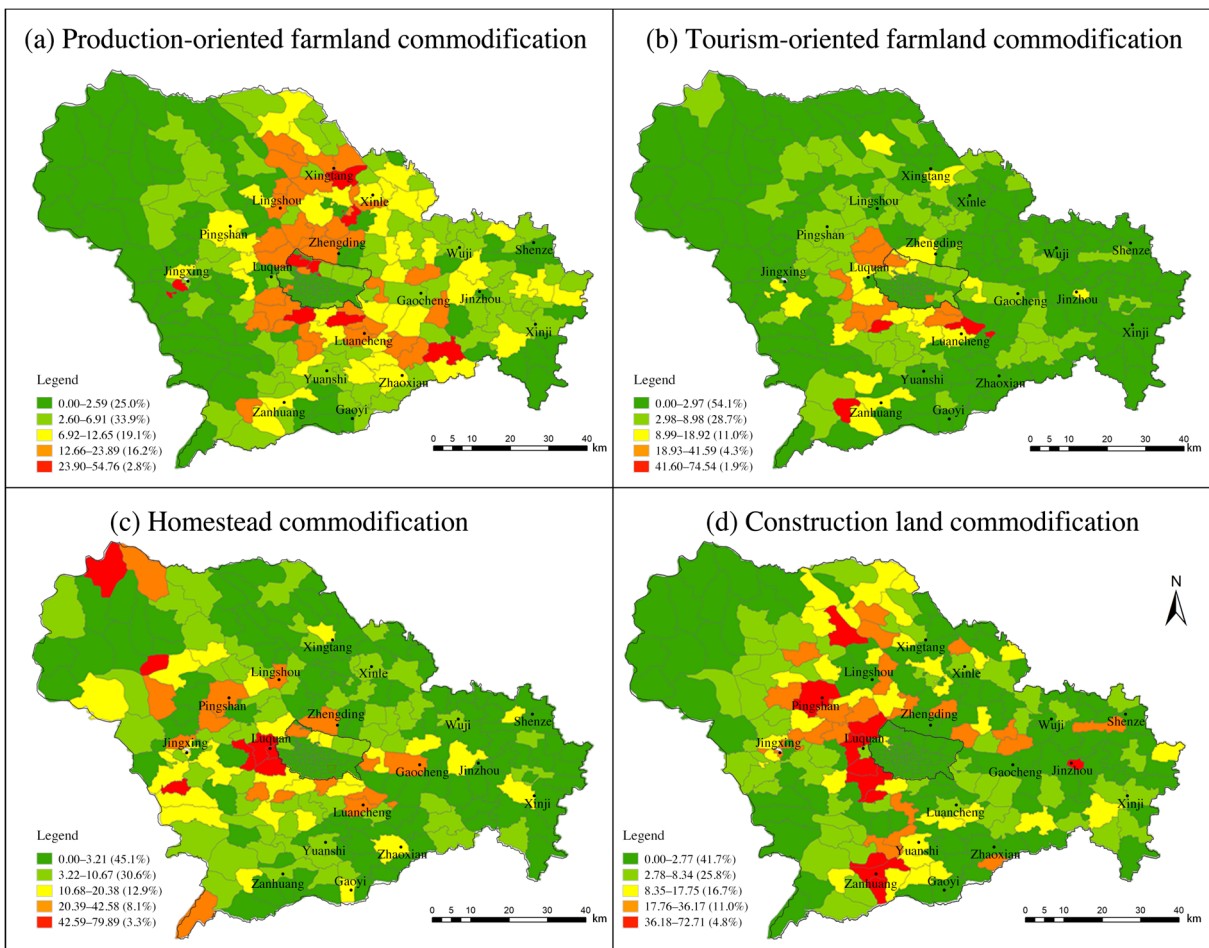

**Figure 3.** Density map for RLC by township scale.

## 4.2. Spatial Autocorrelation of Rural Land Commodification

Through the spatial autocorrelation test, we examined the spatial patterns and differences in the four types of RLC from a spatial statistical perspective. The Global Moran's Index showed a positive spatial autocorrelation for types of POFL, TOFC, and CLC (Table 2), with a positive Moran's Index >0.20 and a *p*-value < 0.01. This result revealed the aggregation pattern of these commodification types at the township scale. In other words, these types of commodification posed a positive connection and interaction between adjacent townships, with townships at a high development level influencing the growth of neighboring townships. By contrast, the spatial autocorrelation of HC was not obvious, with a Moran's Index of 0.09. This result revealed a random distribution pattern of HC.

**Table 2.** Global spatial autocorrelation analysis results.

|  | **POFC** | **TOFC** | **HC** | **CLC** |
|---|---|---|---|---|
| Moran's I | 0.303 | 0.215 | 0.090 | 0.203 |
| Z score | 9.75 | 7.12 | 5.74 | 6.70 |
| *p*-value | 0.001 | 0.001 | 0.001 | 0.001 |

The distribution of spatial clusters can be further visualized with the use of LISA cluster maps. As shown in Figure 4, the cluster distribution presented a significant spatial heterogeneity, and the H-H clusters accounted for the largest proportion of the four

commodification types. In the case of POFC, the H-H clusters were distributed in the northern and eastern parts of Shijiazhuang city, including Hancun, Beiwa, Nanzhai, and Tongye township, etc. These areas covered the production region of two geographical indication products (GI), "Zhaoxian snowflake pear" and "Lingshou enoki mushroom", showing an obvious trend of agricultural modernization and industrialization. In the case of TOFC, the H-H clusters were mainly located in the western hilly areas such as Dahe, Shangzhai, Jicun township, etc. The physical and geographical conditions of fruit farming are superior in the region, and the agricultural structure was dominated by cash crops such as apple, cherry, and Chinese date. As for CLC, the H-H clusters are mostly concentrated surrounding Pingshan, Luquan, and Zanhuang towns, where the population and infrastructure conditions are well developed due to their geographical location. The rural secondary industry developed rapidly under the economic spillover effect from center towns, and thus the rural construction land commodification is obvious in these regions. In summary, the results reveal the diffusion effect of RLC at the township scale.

### 4.3. Influencing Factors of Rural Land Commodification

Based on the spatial analysis above, this study further employed the OLS model to identify the major influencing factors underlying the spatial distribution patterns for each type of RLC. Prior to the construction of models, we tested the variance inflation factor (VIF) of all independent variables and found that all VIFs were less than 4, indicating a reliable result of the model's setting. Table 3 presents the result of multiple regression analysis. Overall, the $R^2$ adjusted values all reach 0.25, showing that the model has a satisfied goodness-of-fit for RLC. Among the four types of RLC, the model for POFC has the best explanatory ability with the highest $R^2$ adjusted value of 0.408.

**Table 3.** Multiple regression analysis of rural land commodification.

| Variables | POFC Coefficient | TOFC Coefficient | HC Coefficient | CLC Coefficient |
|---|---|---|---|---|
| | Population factors | | | |
| Population density | 0.656 * | 0.586 * | 1.955 *** | 2.573 ** |
| Aging population rate | 0.245 | 0.637 * | 1.570 *** | 0.668 |
| | Economy factors | | | |
| GDP per capita | 0.278 | 1.541 *** | 1.309 *** | 3.047 *** |
| Net income per capita | −0.665 ** | −0.058 | −0.837 * | −2.716 *** |
| | Agriculture condition factors | | | |
| Cultivated area | 0.920 *** | 0.719 ** | −0.680 | 2.316 |
| Gross output | 1.201 *** | 0.726 ** | 0.0510 | 0.140 |
| Grain production output | 0.441 | −0.665 * | −0.756 | −1.964 |
| | Location factors | | | |
| Dis to city | −1.691 *** | −1.912 *** | −0.177 | −2.021 ** |
| Dis to county | 0.690 ** | 0.841 ** | 1.153 ** | 0.105 ** |
| Dis to tourist spot | −0.379 | −0.168 | −1.397 *** | −0.149 |
| $R^2$ adjusted | 0.408 | 0.280 | 0.306 | 0.260 |

Note: *, **, and *** indicate values that are significant at the 10%, 5% and 1% levels, respectively.

The OLS analysis revealed the correlation between RLC and various physical and socioeconomic drivers. Regarding population factors, the positive effect of population density was statistically significant on all types of RLC, indicating that human resources play an important role in the land commodification transition. This result supports the argument regarding the relationship between population density and rural livelihood diversity [43]. Due to the limited land resource and large surplus labor, farmers in densely populated areas tend to choose an efficient and high-return land use method, which directly promote the diversification of rural livelihood and industry, especially the development of non-farm activities. TOFC and HC were positively correlated with population aging, implying that the region with a higher number of elderly people is more likely to undergo

the land use transition toward tourism function. Aging townships generally face serious labor shortages and land vacancy problems. Utilizing idle houses and farmland by renting them out as tourism products can greatly relieve the labor and management burden of the rural elderly. However, this influence was insignificant for POFC and CLC, which typically have a substantial demand for labor forces.

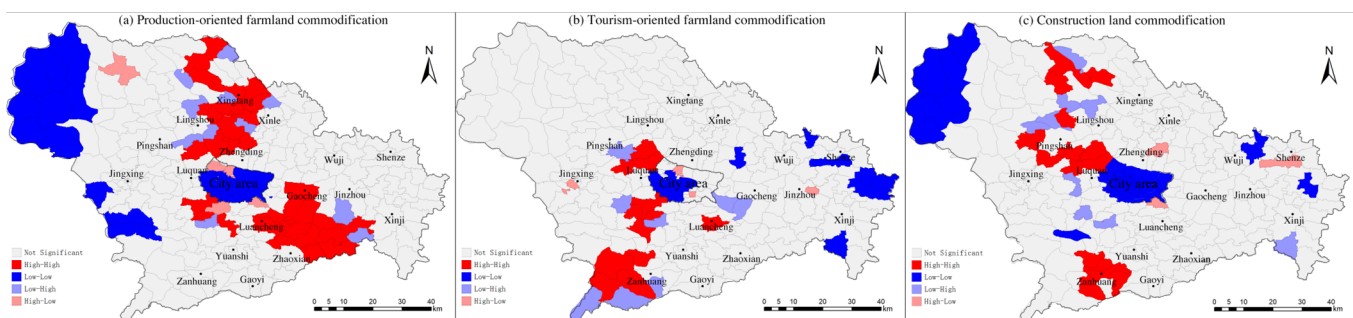

**Figure 4.** LISA cluster map for RLC.

Among economic factors, GDP per capita was positively related to TOFC, HC, and CLC; in other words, these types of commodification were economically sensitive and more prevalent in townships with a higher level of economic development and fiscal capacity. Net income per capita, however, had a significant negative influence on all RLC types except for TOFC. This finding reveals the underlying association between land commodification and rural poverty.

In terms of agricultural conditions, we discovered that cultivated land area and gross output positively influenced POFC and TOFC. That is, farmland commodification was more likely to occur in areas with abundant land resources and high agricultural intensity and efficiency. Grain production output was negatively related to TOFC, confirming the constraining effect of the grain-producing function on diversified coordinated development. In contrast, no significant correlation was discovered among agricultural condition factors, HC, and CLC.

As for location factors, distance to city and to county had similar impacts on POFC, TOFC, and CLC specifically, as these types of commodification increased with proximity to cities but decreased with proximity to counties. The result demonstrates the radiation effect of urban capitals and markets; the closer a township is to the city, the greater radiation and transformation forces it received on agricultural and industrial development elements. Moreover, the negative impact of counties might be related to the rapid urbanization in Chinese small towns, as it commonly results in land expropriation for urban expansion in neighboring townships. Distance to tourist spots has a significantly negative correlation with HC, indicating HC's reliance on tourism resources. This finding could be explained by the fact that rural homestays, the most common type of HC, rely heavily on scenic radiation to attract customers.

## 5. Discussion and Conclusions

The study on the spatial distribution and influencing factors of RLC are vital for revealing the process of rural multifunctional transition. In this study, we analyzed the spatial distribution and the factors affecting RLC with spatial analysis and OLS methods, quantitively revealing the interaction and the driving mechanism from the aspect of population, economy, location, and agricultural conditions. The conclusion and discussion are as follows.

### 5.1. Causes for Spatial Aggregation of RLC

The spatial analysis result shows a positive spatial autocorrelation among POFC, TOFC, and CLC, meaning that those types of RLC present strong aggregation effects at the township level. Similar regional characteristics and interaction between neighboring

regions might provide a general explanation for this result. For one thing, in Chinese rural communities, information exchange and knowledge sharing are common among households, cadres, and local authorities. Once a household or village implements the commodification pathway in their land resource development and achieves competitive advantages in the market, the successful experience spreads quickly through social and administrative networks and inspires widespread imitation in neighboring communities [44]. For another, internal factors such as resource endowment and location conditions affect the spread of the commoditization phenomenon among neighboring townships, creating localized clusters of advantage.

*5.2. Influencing Factors of RLC*

The result of the influencing factor analysis reveals the complex and comprehensive impact of population, economy, agriculture, and location condition on the process of RLC. Because of differences in the mechanisms of commodification models, the factors drive progress to varying degrees. In terms of socio-economy base, townships with a higher population density and GDP generally have a higher development level of RLC. This result corresponds with those of previous studies. For one thing, economy and population play an important role in RLC, for they provide financial and labor support for industry transitions and significantly affect enterprises and government investment. With the continuous inflow of external capital, the level of rural resource development was enhanced and further led to an extensive transition process [45]. For another, RLC also facilitates land appreciation and industrial transformation under the intervention of the market, leading to positive effects on rural economies and employment [46]. Such economic growth and population influx effects have crucial implications for local governments in their land use decision-making. The study further revealed that RLC is more prevalent in regions characterized by a higher aging and lower per capita income. The underlying reasons for this phenomenon may be attributed to the pursuit of economic benefits by local governments and residents through land transfer and trading. Due to the lack of labor and technology, areas with disadvantaged conditions typically rely on land transfer to obtain external financial and technological support, and the basic income obtained from land leasing in turn becomes an important source of income for the rural poor [47]. In recent years, land transfer played a more important role in Chinese rural poverty alleviation. This development strategy resulted in the accumulation of capital in rural areas and the activation of the asset and commodity characteristics of rural land.

Agricultural conditions, such as cultivated areas and output, affect farmland commodification. In the study region, townships with abundant resources, high agricultural intensity, and efficiency boast a higher level of farmland transition for industrialization and tourism. Developed agricultural areas have a wide range of enabling factors and high transitional potential, as they are often large, highly mechanized, and well-capitalized. Their excellent agricultural conditions allow them good opportunities for the implementation of multifunctionality pathways, and farmers in these regions have advantages in conducting agricultural production with higher economic added-value and stronger competitive advantages such as farm tourism and product marketing. These townships can serve as role models for agriculture multifunctional development, promoting the formation of regional industrial clusters and thus driving future farmland transition.

Location condition is fundamental in determining a township's access to resources. Proximity to a city is the key factor for POFC, TOFC, and CLC presence, but for HC to thrive, being close to tourist attractions is more important. When a rural area is close to a developed city, capital, and market demand influence rural land through the effect of urban radiation, resulting in effective land development regarding leisure agriculture, rural processing, and other industries. Similarly, villages near tourist sites are susceptible to tourism development, which leads to the conversion of residential areas to accommodate the excessive demand for hospitality.

### 5.3. Policy Implication

The commodification of rural land can spur rural transformation, providing financial and technological support for the enhancement of consumption value, diversification of the rural economy, and the exchange of urban–rural resource elements. Land commodification still has negative impacts that cannot be ignored. Looking ahead, RLC will remain a key driver and component of Chinese rural transformation. In order to integrate RLC into national strategies for rural revitalization and sustainable development effectively, while mitigating its negative effects, policy suggestions should be proposed.

The existing spatial distribution of RLC is characterized by spatial concentration and polarization. This distribution pattern helps to stimulate the industrial cluster effect and enhances the competitive advantage of the rural industry through high resource integration. However, if unregulated, blind land development can lead to the overproduction of homogeneous products, resulting in competition and erosion of existing resources [48,49]. To prevent this, government intervention is necessary to ensure a balanced and coordinated development. For townships with a high degree of commodification, land development policies or regulations should be implemented to guide the market and prevent unnecessary overlapping construction.

The coexistence of rural land commodification with rural poverty and aging is a cause for alarm. Uleri argues that under the influence of capital markets, RLC might become a source of land insecurity and conflict because of monetary incentives and will increase land scarcity [13]. Furthermore, inadequate land transfer systems can also contribute to the uneven distribution of benefits, leading to social differentiation and conflicts [14]. In the process of rural transition, disadvantaged groups such as the poor and old are generally excluded or play a passive role in rural affairs [50]. Farmers could not share the benefits of land appreciation equally because of information asymmetries but lost their ability to be self-sufficient due to land transfer [51]. The development of rural land transfer in China has been rapid, with a survey conducted in 2021 revealing a national rural land transfer rate of 22%. However, the satisfaction and benefits of farmers have been found to be unsatisfactory [52]. To avoid the further marginalization of vulnerable groups caused by the invasion of commodification, the protection of community decision-making rights and the equitable distribution of benefits are significant.

Financial aid and talent recruitment are the best strategies for rural progress. In recent years, China's targeted poverty alleviation and capital to the countryside strategies have injected significant investments into rural industry transition, which provide an invaluable source of start-up capital for villages with poor conditions and limited development opportunities [53]. However, at the same time, local governments should consider tax incentives and financial support to attract and keep talent, particularly the urban middle class and young entrepreneurs who are crucial for the future rural construction. In addition, against the inevitable backdrop of labor exodus, rural transformation is bound to coexist with population aging in the long term. Therefore, while guiding old farmers out of production and giving way to commodification development, the government also needs to improve the social security system and enhance the basic welfare of farmers, to balance development and farm household livelihood security and avoid the 'new rural poor' because of the loss of land resources.

Various development pathways are possible for rural commodification. Besides farmers' spontaneous land construction, local government also needs to base on the foundation of resources, exploring the development and management system adapted to regional conditions. The first task for regions with advantageous agricultural resources is to speed up farmland consolidation and strengthen village-enterprise cooperation, expanding the scale and branding benefits through innovative agricultural product development. For areas close to tourism and urban source markets, villages should make full use of the radiation advantage and focus on improving the quality of rural spaces, such as the construction of organic farms and B&Bs. In addition, the composite RLC development pathway represented by "agricultural products plus e-commerce" and "rural complex" has proven to

have a wide spectrum of development in China [54,55]. It optimizes the industrial structure, improves the economic and social resilience of the countryside, and is crucial to sustainable rural transformation in the future.

### 5.4. Limitation and Future Work

Although this study could be considered an attempt to explore the RLC phenomenon through the lens of spatial patterns, there are still certain deficiencies that require further investigation in future studies. Firstly, due to data acquisition limitations, this study only addressed the driving mechanism from the perspective of location and social economy. However, the RLC is a complex and diverse system that is also influenced by policy and social values. An in-depth analysis is required from multiple dimensions such as farmer intentions and government initiatives. Second, macroscale pattern analysis is insufficient to capture the developing details of RLC. Therefore, future research should expand the scope of study to typical case studies, exploring the evolution and characteristics of rural land transition at the microscopic level. In addition, for the purpose of statistical analysis, this study categorized various types of RLC patterns as separate components and conducted classified research. However, the multifunctionality of commodification transformation, that is, the synergistic development of different RLC types, is of vital importance for theoretical research on RLC. Future studies should analyze from an integrated and unified perspective, exploring the interrelationships among different types.

**Author Contributions:** L.F., Conceptualization, Methodology, Validation, Analysis, Visualization, and Writing—Original draft preparation; J.S., Supervision and Writing—Reviewing and Editing. All authors have read and agreed to the published version of the manuscript.

**Funding:** This work was supported by the China Scholarship Council under the State Scholarship Fund (grant no. 202208050048).

**Data Availability Statement:** The data involved in this article come from the statistical yearbooks of local authorities, and the author can provide them if requested.

**Conflicts of Interest:** The authors declare no conflict of interest.

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
