# Peer review of "Spatial Patterns and Influencing Factors of Rural Land Commodification at Township Scale: A Case Study in Shijiazhuang City, North China"

_land, doi:10.3390/land12061194_

Round 1

Reviewer 1 Report

The article can be of interest, however the absense of international comparison limits its benefit. The social aspect is also largely missing. More specifically:

Line 33-34: reference is needed

Line 49: coherence in references

Lines 68-70: here is totaly missing the problematisation of this process that has critical effects on a social level; I suggest to take into consideration the abundant literature on the agrarian question. You should definitely referr to Polanyi, K. (2015). The great transformation, and then as suggestion:

- Uleri, F. (2021). Peasantry and Agrarian Capitalism from below: The Peasant Communities of the Bolivian Southern Highlands under the Quinoa-boom. Ager: Revista de estudios sobre despoblación y desarrollo rural= Journal of depopulation and rural development studies, (32), 39-63.
- Bernstein, H. (2006). Is there an agrarian question in the 21st century?. Canadian Journal of Development Studies/Revue canadienne d'études du développement, 27(4), 449-460
- McMichael, P. (1997). Rethinking globalization: the agrarian question revisited. Review of International PoIiticaI Economy, 4(4), 630-662.

Line 330: multifunctionality is crutial to analize the commodification, I suggest to consider better this aspect, emerging now but not included in theoretical framework

Lines 362-364: in my opinion social impact of land commodification is not sufficiently explored and deserves much higher attention

Lines 401-405: which are the sources of these affirmations? Data don´t show this and references are missing, seems to be a simple assumption of authors

Author Response

Response to Reviewer 1 Comments

Point 1: Line 33-34: reference is needed

Response 1: Thanks for your suggestions. we have added more references to support the idea that “rural places are increasingly sites for consumption-oriented activities in many countries.”

Point 2: Line 49: coherence in references

Response 2: We are really sorry for our careless mistake. Thanks for your reminder.

Point 3: Lines 68-70: here is totally missing the problematization of this process that has critical effects on a social level; I suggest taking into consideration the abundant literature on the agrarian question. You should definitely refer to Polanyi, K. (2015). The great transformation, and then as a suggestion:

- Uleri, F. (2021). Peasantry and Agrarian Capitalism from below: The Peasant Communities of the Bolivian Southern Highlands under the Quinoa-boom. Ager: Revista de estudios sobre despoblación y desarrollo rural Journal of depopulation and rural development studies, (32), 39-63.
- Bernstein, H. (2006). Is there an agrarian question in the 21st century?. Canadian Journal of Development Studies/Revue canadienne d'études du développement, 27(4), 449-460
- McMichael, P. (1997). Rethinking globalization: the agrarian question revisited. Review of International PoIiticaI Economy, 4(4), 630-662.

Response 3: We think this is an excellent suggestion. As you could find in the line 35-50 of this manuscript, a literature review about land commodification and its critical effects was added.

Point 4: Line 330: multifunctionality is crucial to analyze commodification, I suggest considering better this aspect, emerging now but not included in the theoretical framework

Response 4: We highly appreciate this suggestion. However, it may be challenging to explore the multifunctionality of rural land commodification (RLC) within the current research framework. Your suggestion has been added to the limitations and future research section of the article (line467-470), and we will continue to study this issue with keen interest.

Point 5: Lines 362-364: in my opinion, social impact of land commodification is not sufficiently explored and deserves much higher attention

Response 5: Thanks for your suggestion. As shown in sections 5.2 and 5.3, we separated the discussion of influencing factors and policy implications and added a paragraph (lines 415-428) to analyze the potential crises arising from the coexistence of rural land commodification with poverty and aging communities, as well as some possible suggestions.

Point 6: Lines 401-405: which are the sources of these affirmations? Data don´t show this and references are missing, seems to be a simple assumption of the authors

Response 6: We have added a reference regarding the government guideline for social capital investment in agriculture and rural development to prove this idea (line 433). It can be observed that the key industries and sectors encouraged for investment by the Chinese government are strongly linked to the commodification of rural areas.

Reviewer 2 Report

The core of this article is spatial patterns and influencing factors of rural land commodification. In China, rural land is owned by the state or peasant collectives, and land use is governed by land use general planning.

In Line 54, the authors also mention that previous studies generally revealed the driving mechanism from the perspective of social force and government interference, which is in line with Chinese conditions. In this article, population, economic level, agriculture, and location conditions are identified as key drivers. Is it the economic level that drives the rural land commodification, or is it the rural land commodification that promotes economic development?

In addition, only 28 previous articles are referenced in this article, and only 8 references in the last five years, and no references are cited in the discussion section. The significance and insights of the article are not persuasive enough.

 Minor editing of English language required.

Author Response

Response to Reviewer 2 Comments

Point 1: The core of this article is spatial patterns and influencing factors of rural land commodification. In China, rural land is owned by the state or peasant collectives, and land use is governed by land use general planning.

 In Line 54, the authors also mention that previous studies generally revealed the driving mechanism from the perspective of social force and government interference, which is in line with Chinese conditions. In this article, population, economic level, agriculture, and location conditions are identified as key drivers. Is it the economic level that drives rural land commodification, or is it the rural land commodification that promotes economic development?

Response 1: Thanks for your comments. We highly acknowledge the crucial role of government and policy guidance play in the commodification of rural land in China. However, with the reform of rural land policies, the influence of the market and other external factors on rural land utilization is gradually expanding. (This viewpoint is also supported by this paper: Long, Hualou, et al. "Multifunctional rural development in China: Pattern, process and mechanism." Habitat International 121 (2022): 102530.) The innovation of this article lies in providing a geographical perspective to explain the patterns and causes of rural land commodification.

For the question of the causal relationship between economic development and land commodification. The article utilizes a multiple regression model to investigate the correlation between rural commodification and economic development (namely that commodification tends to occur commonly in regions with better economic development) rather than a causal relationship. We delated the ambiguous section in the paper and added a more comprehensive explanation according to previous research.(line 361-367)

Point 2: In addition, only 28 previous articles are referenced in this article, and only 8 references in the last five years, and no references are cited in the discussion section. The significance and insights of the article are not persuasive enough.

Response 2:  We sincerely appreciate your valuable comments. In the revised manuscript, we have increased the number of references to 55 and paid significant attention to the reliability of the literature. A total of 22 references in the past five years have been cited. You could find this citation in lines 34, 38, 42, 43, 45, 48, 351, 365, 367, 375, 411, 418, 420, 421, 423, 426, 433, and 452.

Round 2

Reviewer 1 Report

Dear authors, thank you for having considered carefully my suggestions. Even if I still disagree with your approach to development , especially when you write "RLC has become an unavoidable trend in many countries" line 46, I think that your article can help inproving knowledge about the process of commodification of land in China. As post-growth researcher, I suggest you to be less optimistic about modernisation and think about your alleged underdevelopment as a resource. This is just a political consideration, nothing to do with your research. Good luck!

Reviewer 2 Report

The article can be accepted in present form.